# Occupational Safety and Work-Related Injury Control Efforts in Qatar: Lessons Learned from a Rapidly Developing Economy

**DOI:** 10.3390/ijerph17186906

**Published:** 2020-09-21

**Authors:** Rafael J. Consunji, Amber Mehmood, Nazia Hirani, Ayman El-Menyar, Aisha Abeid, Adnan A. Hyder, Hassan Al-Thani, Ruben Peralta

**Affiliations:** 1Hamad Injury Prevention Program, Hamad Trauma Centre, Department of Surgery, Hamad General Hospital, Hamad Medical Corporation, Doha 3050, Qatar; nazia.gadhia@gmail.com (N.H.); AELMENYAR@hamad.qa (A.E.-M.); AAbeid@hamad.qa (A.A.); Halthani@hamad.qa (H.A.-T.); rperaltamd@gmail.com (R.P.); 2International Injury Research Unit, Department of International Health, Bloomberg School of Public Health, Johns Hopkins University, Baltimore, MD 21205, USA; amehmoo2@jhu.edu; 3Milken Institute School of Public Health, The George Washington University, Washington, DC 20052, USA; hydera1@email.gwu.edu

**Keywords:** work-related injury, occupational injuries, Qatar, injury prevention, migrant workers, Middle East

## Abstract

Work-related injury (WRI) control is an integral part of occupational safety. In rapidly developing Gulf countries such as Qatar with a predominantly expatriate workforce, WRI control is a complex issue often seen in conjunction with the implementation of labour laws and labour rights. We aimed to implement a public health approach to facilitate efforts to achieve long-term WRI control in Qatar. A range of initiatives helped to gain visibility and momentum for this important public health problem, including identifying and engaging with key stakeholders, workers’ surveys, steps to establish a unified injury database, and the implementation of a WRI identification tool in the electronic medical records. A contemporaneous improved enforcement of existent occupational safety regulations through heightened worksite inspections and efforts to improve living conditions for migrant workers also took place. WRIs are not only a Qatar-specific problem; the same issues are faced by neighbouring Gulf countries and other rapidly developing economies with large expatriate worker populations. These strategies are also useful starting points for similar countries interested in nurturing a safe, healthy and productive workforce.

## 1. Introduction

Qatar is a rapidly developing high-income country in the Middle East with a diverse expatriate worker population that is increasing as the road infrastructure and construction projects ramp up for the Football World Cup (FIFA) 2022 [1,2]. Like other countries in the Gulf Cooperation Council (GCC), Qatar has a large expatriate population, the majority of which originate from South Asia (India, Nepal, Pakistan, Bangladesh and Sri Lanka), the Philippines and Egypt [3,4,5].

For the past few years, labour advocacy groups have raised concerns about the increasing number of deaths and disabilities resulting from work-related injuries (WRIs) in Qatar [6,7,8]. The outcomes and costs of WRIs from falls (from height) and falling objects have been reported in hospital-based publications from Qatar [9,10,11]. Despite other authors highlighting the epidemiology of WRIs and notably identifying the disproportionate risk of expatriate workers being injured in construction and road traffic injuries, there is a lack of sources reliably providing the true burden of WRIs in Qatar [12,13]. Previous research has only analysed data on moderate-to-severe WRIs seen and treated at the national trauma centre [9,10,11,12,13]; data on mild-to-moderate WRIs as well as pre-hospital WRI deaths had not previously been analysed. This evidence gap had become an opportunity for speculation as well as a challenge for implementing focused and locally appropriate occupational safety programs for all types of WRIs, addressed to the populations at the highest risk for such.

There have been considerable efforts to bridge this gap in the last few years. This article aims to (1) provide an overview of the WRI problem in Qatar, (2) describe the steps taken to identify key gaps in describing the true outcomes of WRIs and current efforts towards addressing these gaps and strategies to promote an evidence-based approach towards WRI control and (3) summarize important lessons learnt during the process; this project is a component of the “Unified Registry for Workplace Injury Prevention in Qatar (WURQ)” grant (National Priorities Research Program (NPRP) 7-1120-3-288) awarded as part of the National Priorities Research Program of the Qatar Foundation.

## 2. Landscape Analysis of Work-Related Injury Control and Occupational Safety in Qatar

To understand the rationale for a comprehensive WRI-control agenda, a number of contextual factors must be considered. These factors include, but are not limited to, the WRI burden, current policies and regulations covering WRIs in Qatar, rising demand for WRI data, and inadequate infrastructure for WRI surveillance. These important issues are summarized below.

### 2.1. Burden

A number of analyses have described the magnitude of WRIs in Qatar. Depending upon the mechanism of injury and age group under study, 85–97% of WRI victims are non-Qatari or expatriate [10,11,14]. With respect to the impact on the country’s health care services, it was widely acknowledged that WRIs were not only a major cause of patient admission but also increasing in magnitude and severity. It has been reported that approximately 30% of all trauma admissions are due to WRIs, with the most common causes being falls from heights, being struck by a falling object and suffering from a road traffic injury [12]. Pedestrian injuries were reported as a significant cause of WRI and trauma admission in Qatar, and laborers from South Asia were noted to be at disproportionate risk for this type of injury [11,15]. Head, face and neck injuries are fairly prevalent with industrial workers, whereas road injury victims commonly sustain musculoskeletal injuries [14,16]. The population WRI fatality rate is 3.5 per 100,000, which is a conservative estimate owing to the paucity of data, whereas the hospital-based case fatality rate is reported to be 3.7%, which is higher for fall injuries [12,17]. As expected, young, male, migrant workers bear most of the WRI brunt, stemming from difficult, dirty and dangerous (the three Ds) jobs [3,14,18]. Most of these published studies are from the national trauma centre registry data and hence exclude minor injuries treated in out-patient settings and victims who succumb to their injuries in the pre-hospital setting.

### 2.2. Existing Policy and Regulations about WRIs

The Qatar Labour Law defines the minimum standard of rights and benefits for employees to which employers must adhere as well as the obligations of employees working in Qatar [6]. These laws cover employer–employee contracts and regulations, the Kafala system for migrant workers, unions, sponsorship and wage protection system, among many other aspects [19]. Women are discouraged from being employed in arduous or dangerous work, as well as jobs that are detrimental to their health or morals. The Qatar Labour Laws requires employers to protect the workers from injuries, disease and accidents and provide personal protective equipment, first aid boxes, access to on-site nurses and physicians, and periodic medical examinations [20]. The employers must also assume responsibility for medical expenses for work-related injuries [21].

Employer-based or private health insurance is not popular in Qatar; instead, the Qatari government provides all legal residents with free emergency health care in conjunction with the Hamad Medical Corporation (HMC), which is the largest health care provider organization operated by the government. Any worker who sustains a WRI is entitled to receive their full wage during the treatment period or for up to six months. WRIs resulting in death or permanent disabilities are also liable for compensation, with certain exemptions (e.g., proven self-harm, and use of drugs or alcohol). Additionally, there is a need to improve workers’ access to and knowledge of policies, procedures and public documents produced by relevant ministries, as well as interpretation and translation services, considering the multilingual nature of the workforce in Qatar [3,22].

### 2.3. Spotlight on WRIs as a Public Health Issue in Qatar

Since the boom of construction activities in the wake of FIFA 2022 projects and a rapid influx of the migrant labour workforce, the need for WRI prevention and control has been heightened. This was partially seen in the context of labour recruitment and labour laws, some of which were deemed less than ideal by international organizations such as Amnesty International. The Qatari government generally denies the veracity of such claims and has responded by improving Qatari Labour Laws in a series of amendments [23], providing improved access to health services for all documented migrant laborers and also placing WRIs as a research priority issue in the National Priorities Research Program (NPRP) of the Qatar National Research Foundation [24]. These changes and reforms are still far from ideal, and there are concerns surrounding unjust working conditions, the treatment of migrant workers and the repatriation of migrant workers without notice after injuries [25,26].

### 2.4. Rising Demand for Reliable WRI Data

As the population of Qatar is rapidly growing, most notably, with the 50% increase in the labour force from 2007 to 2012 (2012 Labour Force Survey), the need and demand for reliable WRI data has also increased. Only the Hamad Medical Corporation (HMC), the national health care provider, was capable of fulfilling bi-annual data requests from the local government ministries for data captured in its trauma registry [12,14]. It was also noted that the scope and process of WRI data collection did not match the requirements of the ministries for using this information and “responding” to trends or conducting inspections on hazardous worksites. Additionally, oftentimes, these did not include data elements that were deemed important for WRI control and economic burden, i.e., workplace conditions, compensation or days off from work. Another important area that was affected by this lack of evidence was the identification of policies, laws, mechanisms, industries and vulnerable populations for injury prevention and occupational safety.

### 2.5. Inadequate Infrastructure for WRI Data Collection

There is no official framework for data sharing or communication between the different stakeholders for Occupational Health and Safety (OHS) in Qatar, leaving many stakeholders unaware of the content and frequency of the WRI data collected by the other agencies. Certain large government and private entities that employ thousands of laborers, such as the Public Works Authority (Ashghal), petrochemical companies, FIFA 2022 projects, Qatar Rail and Hamad Medical Corporation, appear not to have any WRI-reporting relationships with either the Ministry of Labour or Pubic Health. Additionally, there is no gold standard for data collection on WRIs on a nationwide basis; only industry-specific examples were found (i.e., petrochemical, construction and farm work) [27]. This has resulted in a fragmented, incomplete and inaccurate WRI data collection system in Qatar and paved the way for speculation from external parties. This situation also adds to the lack of mechanisms to corroborate and validate WRI statistics presented by different sources.

## 3. Multisectoral Efforts for Advancing WRI Prevention and Control in Qatar

Creating safe workplaces that are responsive to the needs of the labour force is the ultimate goal of the Qatari authorities. Promoting an evidence-based approach to WRI control in Qatar is constantly hampered by the dearth of reliable data to provide a full and factual picture of this important public health problem. Core components of WRI surveillance such as the measurement of burden, risk factors, the effectiveness of different interventions and outcomes cannot be addressed through a single-centre trauma registry. From this arose an urgent need to create a concerted effort to finalize a data collection and sharing framework, developing a consensus on the data elements essential for different stakeholders, and creating a platform where WRI-control policies and strategies could be proposed and deliberated for further action. This NPRP grant (Work-related Injuries Unified Registry in Qatar: WURQ) was launched in 2015 to identify, document and address these aforementioned gaps. This multi-pronged effort, its processes and its outcomes, starting with Item 1.5 in the previous section, that were conducted within the context of the NPRP grant are described below.

### 3.1. Stakeholder Engagement

WRI prevention and control is a multi-sectoral, interdisciplinary effort that can only be achieved through the identification, engagement and active participation of all the relevant stakeholders. Essential to this process is to ensure the engagement of the workers and Occupational Safety and Health (OSH) specialists for training and education. Similarly, the health sector actively contributes towards injury prevention programs as well as the provision of timely and appropriate care to WRI patients.

Initial stakeholder mapping was followed by formal engagement using multiple strategies. Meetings with key employees and staff were held to highlight the importance of WRI surveillance in the Qatari context and explain the objectives of the project. The stakeholders and their roles are outlined in Table 1 below.

These engagements were used to seek permission to access their WRI data and invite them to multi-sectoral meetings with the following objectives: (1) the standardization of terminology and determination of minimum dataset requirements; (2) eliciting input and suggestions for improving WRI data in Qatar, including mechanisms for data sharing between different institutions; (3) providing a platform for all stakeholders to better understand reporting relationship/s and statutory requirements to submit/collect reports; (4) the identification of additional data sources and Occupational Safety and Health (OSH) stakeholders; and (5) the discussion of avenues for the implementation of injury prevention programs, training and education.

As these engagements helped in creating a stakeholder network, a series of meetings were conducted in order to discuss and develop consensus on an analytic framework (Figure 1) for the development and implementation of policies, data sharing and coordination for educational programs and health services to improve worker safety in Qatar [22].

A unified WRI database was recognized as a much-needed core foundation for evidence-based programs, policies and further research.

### 3.2. Identification and Assessment of Select Data Sources

An initial mapping of the existent data sources was performed to gain access to and examine the nature and quality of the WRI information. This process was further aided by mapping the flow of WRI patients in Qatar to identify data sources from health and non-health entities (Figure 2), such as the Trauma Registry, Emergency Departments, Mortuary, Ambulance Service and Rehabilitation Department of the HMC. These databases were evaluated for essential WRI data elements, database format, timeliness, reporting requirements and quality of data. The injured worker (Row 1) is triaged at the first point of clinical care and data capture (Row 2), according to the severity of injury. The 3rd row represents additional data capture points for patients admitted to the HMC or private hospitals and WRI fatalities. The 4th row represents data capture points of discharged patients. The blue boxes represent data sources included in the unified registry. The green boxes represent data sources that are not captured by the unified registry.

The main finding from this analysis was that there was no consistent system for coding an injury as work- or not work-related, except for the Hamad Medical Centre trauma registry, which had been compliant with the standards of the American College of Surgeons Trauma Quality Improvement Program (ACS-TQIP) and the ACS National Trauma Data Bank (ACS-NTDB) since 2010 and therefore had a mandatory data field that indicated the work relatedness of the injury. These initial findings were validated by an analysis of a 1-month sample from each of the data sources. The initial assessment of the inconsistency and variability of the data was demonstrated and confirmed [27]. Other important shortcomings of most data sources included the non-uniformity of case definitions, the coding systems, incomplete injury details, and a lack of documentation of the clinical and social outcomes for individual cases.

### 3.3. Standardization of Terminology and Minimum Dataset

The findings from the data source evaluation were presented in a multi-sectoral stakeholder meeting to integrate and standardize data sources [27]. The discussion led to a number of consensus points such as a standard definition of a WRI using International Labour Organization (ILO) standards [28] (Table 2).

### 3.4. WRI Patients and Workers’ Survey

Two separate surveys were conducted to inform the development and implementation of targeted interventions to improve worker safety, health-seeking behaviours and subsequent re-employment or deployment. A hospital-based survey was conducted to explore the work circumstances and environments leading to WRIs requiring inpatient treatment [29]. The survey revealed that the majority of the workers deemed WRIs as “accidental”. Of them, 58% had some form of safety training and 78% used some form of personal protective gear. Only half of them said that they had some form of medical insurance. Similarly, another survey was conducted during the “Health at Work” fair organized by the Ministry of Public Health (MoPH; formerly known as the Supreme Council of Health) on World Day for Safety, 2015, capturing workers from various industries. The survey revealed that at least 6.4% had had a WRI in the preceding 12–24 months. As documented in the previous studies, injuries in the construction industry were the most common (59%), followed by those in water supply (11.8%). Falls, being struck by falling objects and machinery-related injuries were the most common mechanisms of injury. Most of the injured workers sought care from the national health care provider, the Hamad Medical Corporation, while almost a quarter needed hospital admission [30].

### 3.5. Introduction of WRI Identification Mechanism in Electronic Health Records

The project team worked with the information technology teams of the HMC and the Qatar Red Crescent to include a separate question and checkbox for identifying each WRI in all electronic health records regardless of in-patient treatment or presented complaints. The opportunity to introduce a mandatory WRI field into the electronic health record systems was utilized to expand the coverage of WRI data in Qatar. To test the reliability and validity of the mandatory WRI mechanism, data collection for two 1-month periods was conducted, 6 months apart, to ensure the reliable capture of all the WRIs seen in the emergency departments and outpatient settings. This led to the further sensitization and training of staff designated to collect WRI information. The implementation of this simple-but-effective method for documenting WRIs in the emergency and urgent care settings of both government and private health care facilities coupled with standard reporting guidelines may reduce the under-reporting of minor injuries. A wider implementation of this WRI checkbox is planned, to expand it to other health institutions and outpatient clinics that treat workers with WRIs. This registry is conducting a prospective collection of WRI data, and the results and analysis of these data will be the subject of future research from the WURQ team.

## 4. Discussion

This paper details the diverse and multisectoral efforts taken to address work-related injuries in Qatar using a public health approach, supported by a National Research Priorities Program. These ongoing efforts employed strategies to develop systems to accurately and completely describe WRIs; applied globally accepted WRI definitions, metrics and indicators; increased the awareness and evidence about WRIs and their prevention; brought OSH stakeholders together; provided a platform for regular and sustained interaction on key issues; created an environment for the integration of various data sources; and developed reliable measures for documenting minor and moderate WRIs seen and treated outside of the national trauma centre. This is an ongoing process that was faced with many challenges, but continuous and concerted efforts and a multi-pronged strategy are crucial for sustainable results. Several lessons learnt during the process are shared/summarized here.

### 4.1. Stakeholders’ Engagement Is the Key

Framing the issue in a manner that resonates with all involved parties increases the likelihood of getting buy-in and bringing stakeholders together for consensus development. As Qatar is a rapidly developing country with job opportunities in almost every industry, the expatriate population is growing and there are a number of stakeholders that are engaged in recruitment, employment, training and education and implementing workers’ safety programs. As much as WRIs adversely affect the health of the workers, with possible negative outcomes such as disability or death, they are also a burden on the health services. Additionally, the Ministries of Public Health and Labour, Ashghal, are directly impacted by this issue that not only results in lost productivity but also costs a sizeable proportion of GDP for treatment, recruitment, employee replacement and training, insurance claims and repatriation.

Listening to and providing an opportunity for workers, as key stakeholders, to share their WRI experiences and raise concerns about their own institutional preparedness and safety practices, as demonstrated in both of our surveys, helps in the planning of actionable and locally appropriate targets for future interventions. Internal bureaucracy as well as the pressure to respond to international criticism might lead to short-term, irrelevant or ineffective solutions. Recognizing early on that the process is long and requires persistent effort helps in the continued “sensitization” of key players for maintaining the momentum and pushing the agenda forward.

### 4.2. Reliable WRI Data Are the Single Most Important Investment for Evidence-Based Policymaking

While the international criticism about the rising WRI burden and inadequate safety practices in Qatar’s construction industry was making headlines, the main shortcoming that prevented an informed and evidence-based response was the lack of integrated and reliable data documenting the WRI burden. Although the epidemiological studies published from the HMC trauma registry data provided a solid foundation for assessing the health service burden and identifying vulnerable groups, they were not enough to describe the entire picture of the WRI burden in Qatar [3,9,10,16]. The multiple approaches conducted in the context of a multi-phased research project not only made inroads into the initial assessment of existing data sources but also created a consensus on a definition and minimum dataset for WRIs, primary data collection from workers and WRI victims about their safety practices, and risk mitigation in the workplace. Furthermore, a simple mechanism such as the introduction of a WRI identification tool (e.g., mandatory checkbox) in the EMR system enabled the capture of moderate-to-minor injuries that are largely left out of trauma registry and hospital admission logs. Similar efforts should be directed towards creating data-sharing guidelines for minimum data requirements and policies for mandatory data submission for all industries to facilitate accurate statistics and summary reports about WRI incidence, near-miss incidents or harmful exposures, the demographic characteristics of the injured workers, recidivism, days away from work, and fatalities, similar to the factsheets and reports available in other countries and industries [31,32,33,34]. This, in itself, could be an incentive for the employers and employees to improve workplace safety, more so if these are supported by the imposition and enforcement of laws that mandate their inclusion in all employee databases.

### 4.3. Policy Framework and Adoption of Public Health Approaches for WRI Control

One of the most critical steps for achieving long-term and sustained WRI control is to develop integrated and broad-based policies taking a public health approach. To date, WRIs have been highlighted as an industrial safety and human rights issue. As described in our earlier publication [22], efforts directed towards applying a public health approach to WRIs that integrates policy, oversight and research to inform practice, interventions and training will ensure long-term injury control and the physical and mental wellbeing of workers from all backgrounds, industries and occupations (Figure 1). Hazard mitigation, near-miss incident analysis, and events reports must be used to improve occupational safety standards. The education and training of the employees and promotion of safe behaviour at work requires the simultaneous engagement, encouragement and incentivization of employers, with oversight and enforcement from the government regulatory bodies. Creating a formal body or government-mandated committee, where injury control experts, occupational and industrial safety professionals and policymakers could work with the grassroot implementers is critical for realizing this goal.

### 4.4. Dissemination of Findings to a Larger Audience

Continued education, orientation and advocacy within the OSH stakeholder community will improve WRI knowledge and situational awareness and sensitize them regarding the gaps and opportunities in the field of OHS in Qatar. Knowledge dissemination should be supported by up-to-date WRI facts and figures. However, that will only be effective when tailored to a specific target audience, using appropriate platforms to attract different end-users, age groups and educational backgrounds. The diversity of Qatar’s workforce could possibly be seen as a challenge, but it also presents an opportunity for innovative and creative means to test a variety of different approaches for knowledge dissemination, information sharing and public education in community and hospital settings, such as social media engagement, chatbots, multimedia advertising and data incentives [35,36,37].

### 4.5. Limitations

While this project has gone beyond the confines of hospital-based data collection and analysis, the authors acknowledge that it was unable to fully describe the entirety of the occupational safety landscape of Qatar. Data sources that do not have a legal requirement to report WRIs to government ministries, such as the petrochemical corporations and private clinics or hospitals, were not included in this project. While the workers in these entities may represent 10–12% of the population at risk for WRIs, those in these industries or with WRIs seeking care at private clinics may not be representative of the highest-risk industries or worker populations. The newly opened ILO country office was not a part of the initial stakeholder engagement, but efforts are underway to fully engage them in future WRI-control activities.

Mental health and psychological risk and protective factors for WRIs have been reported in different migrant or expatriate worker populations [38,39,40]. A recent review reported that one in four publications on international migrant worker health was in the “mental and psychosocial health” domain [41]. Local data sources that captured risk factors in these domains were not identified but should be the focus of future work in this field. Lastly, the authors acknowledge that there are many challenges to implementing these processes in any setting and that they will be institution- or setting-specific, with even more unique solutions. However, a thorough discussion of these challenges and solutions must be conducted within one’s local context and is beyond the scope of this paper.

## 5. Conclusions and Next Steps

This paper describes the processes that have taken place, within the context of a National Priorities Research Program grant, to address identified gaps in the occupational safety landscape in Qatar. The sustained implementation of a prospective, accurate and complete WRI data collection system through an integrated/unified WRI registry that is fully staffed and funded was identified as the number one priority. This grant also included efforts to convene stakeholders, inventory all sources of WRI data, arrive at a consensus regarding WRI definitions, describe WRI risk factors and disseminate these findings to stakeholders. WRIs are not just a Qatar-specific problem—the same issues are faced by neighbouring GCC countries and other rapidly developing economies with large expatriate worker populations. Finding solutions that are applicable to similar settings with comparable occupational and demographic profiles will help in reaching a vision that is shared by most countries: a safe, healthy and productive workforce. Future endeavours should include studies focused on high-risk populations and industries that evaluate locally appropriate and proven interventions for enhancing industrial safety.

## Figures and Tables

**Figure 1 ijerph-17-06906-f001:**
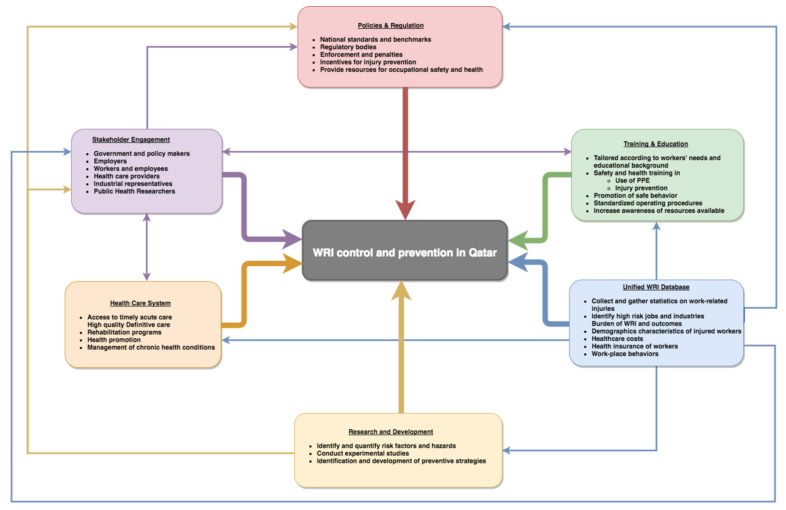
A comprehensive framework for WRI prevention and control in Qatar (reproduced with the permission [22]).

**Figure 2 ijerph-17-06906-f002:**
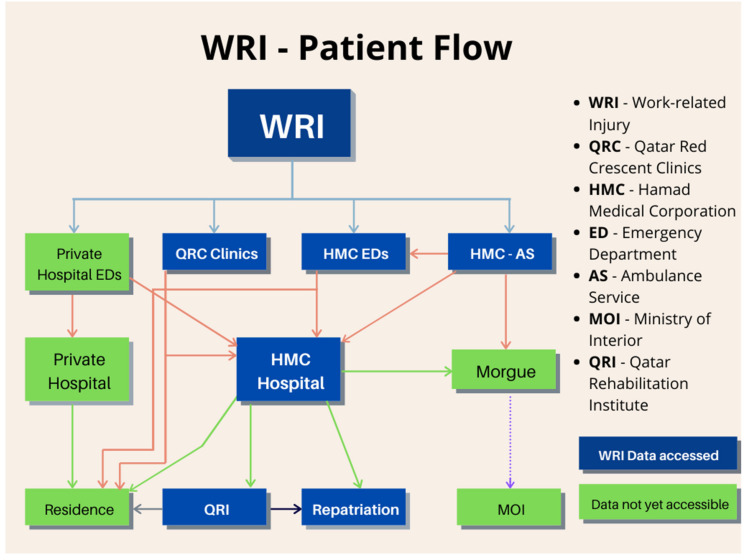
Mapping of the existent data sources and flow of patients with WRIs in Qatar.

**Table 1 ijerph-17-06906-t001:** Stakeholders’ role in work-related injuries (WRIs), Qatar.

Key Stakeholders	Role in WRI Surveillance and Occupational Safety
Ministry of Administrative Development, Labour and Social Affairs	Labour recruitment and oversight
Ministry of Public Health	Official public health authority
Public Works Authority (Ashghal)	Official agency providing regulatory oversight of private companies in construction sector
Qatar Red Crescent	Non-govt. health care provider serving expatriate workers with minor and moderate illnesses and injuries
Deutsche Gesellschaft für Internationale Zusammenarbeit GIZ, Qatar	International public health consulting agency; advisors to the government on occupational safety standards and regulations
Qatar Petroleum	Provides regulatory oversight of private companies in the oil and gas sector
Sidra Hospital	Non-govt. health care provider with a designated role as a provider of health care for women and children
Ambulance Service, HMC	Govt.-run ambulance service providing transport to all moderately or severely injured patients
Qatar Rehabilitation Institute, HMC	Govt. health sector; provides rehabilitation services to injured and other patients in need
Mortuary, HMC	Govt.-run service, an important source of information on WRI mortalities
Hamad Trauma Registry, HMC	Trauma registry in a govt.-owned tertiary care trauma centre; main source of WRI recording the burden of severe injuries
Emergency Medicine Department, Hamad General Hospital, HMC	Govt. health sector; common referral site for all acute and critical injuries requiring multidisciplinary or complex care

**Table 2 ijerph-17-06906-t002:** Operational WRI definitions—source: International Labour Organization (ILO) [28].

An occupational injury is defined as any personal injury, disease or death resulting from an occupational accident
An occupational accident is an unexpected and unplanned occurrence, including acts of violence, arising out of or in connection with work that results in one or more workers incurring a personal injury, disease or death.
A case of occupational injury is a case of one worker incurring an occupational injury as a result of one occupational accident. An occupational injury could be fatal (as a result of occupational accidents and where death occurred within one year of the day of the accident) or non-fatal, with lost work time

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
