# Peer review of "Occupational Safety and Work-Related Injury Control Efforts in Qatar: Lessons Learned from a Rapidly Developing Economy"

_ijerph, 2020, doi:10.3390/ijerph17186906_

Round 1
Reviewer 1 Report
In summary, the conclusion must answer the objective.
Figure 2 is unreadable.
It is necessary to review the description of the study, as it appears to be a case report or an experience report.
The study has a relevant theme, however there are weaknesses in its description:
The first paragraph of the introduction is not directly related to the object of study and is not linked with others in the sequence.
The knowledge gap indicated does not support the objectives of the study.
Carefully review, as the description of actions in the country is long-winded.
The illegible figures in the body of the paper need to be revised.
The writing of section “2. Multisectoral efforts for advancing WRI prevention and control in Qatar: ” at certain points it seems to be a description of what is done in the country in terms of public policies and actions. However, interpretations of the authors appear among the results, for example: This goal could be achieved through an ecosystem consisting of health awareness among employers and employees, safe infrastructure, and implementation of policies ensuring prevention and care of injuries.
In general, I believe that many countries face this problem and can benefit from this article, as a synthesis to initiate actions to protect workers' health. Thus, including the challenges for the implementation of these steps in the discussion is essential, as well as detailing how these practices can be improved.
Author Response
Dear Reviewer 1,
Thank you for your thoughtful recommendations and constructive feedback, they are much appreciated and all noted.
We provide a point by point response to your review below:
Comments and Suggestions for Authors
- In summary, the conclusion must answer the objective.
- Duly noted. We have revised the conclusion so it is more closely aligned with the objectives of the paper.
- Figure 2 is unreadable.
- We have re-submitted the figure in a higher resolution.
- It is necessary to review the description of the study, as it appears to be a case report or an experience report.
- We agree, as a matter of fact, we have referred to this paper, internally, as our ‘process paper’. We have revised the initial description of the paper to reflect that it is a description of the steps and efforts taken by the grant as well as the lessons learned.
- The study has a relevant theme, however there are weaknesses in its description:
- The first paragraph of the introduction is not directly related to the object of study and is not linked with others in the sequence.
- We realize that many readers will not be well-versed or conversant with the occupational safety and demographic context of Qatar so we included this abbreviated description in the introduction. If the Editorial team feels it is superfluous then we will remove it from the manuscript.
- The knowledge gap indicated does not support the objectives of the study.
- We have revised our objectives to make this connection more explicit.
- Carefully review, as the description of actions in the country is long-winded.
- We have worked, within the word count constraints of the Journal, to provide an accurate and complete description of the ‘experience’ or process of our efforts. Section 1 provides in-depth context while Section 2 includes a description of the process as well as the lessons learned. We have further shortened the sections that describe the in-country actions, as recommended.
- The illegible figures in the body of the paper need to be revised.
- Duly noted, we have provided higher resolution figures.
- The writing of section “2. Multisectoral efforts for advancing WRI prevention and control in Qatar: ” at certain points it seems to be a description of what is done in the country in terms of public policies and actions. However, interpretations of the authors appear among the results, for example: This goal could be achieved through an ecosystem consisting of health awareness among employers and employees, safe infrastructure, and implementation of policies ensuring prevention and care of injuries.
- Duly noted and edited, as recommended.
- In general, I believe that many countries face this problem and can benefit from this article, as a synthesis to initiate actions to protect workers' health. Thus, including the challenges for the implementation of these steps in the discussion is essential, as well as detailing how these practices can be improved.
- Duly noted, we have added a ‘challenges’ section in the Section 3.5, as recommended.
- The first paragraph of the introduction is not directly related to the object of study and is not linked with others in the sequence.
- Note: I could not see Figure 1 as it was very blurry, so cannot provide any comments related to this figure.
- Duly noted, we have provided higher resolution figures.
Best regards,
WURQ Team

Reviewer 2 Report
The authors describe ongoing efforts made in Qatar to address increasing work-related injuries and what was learned from implementing these efforts. The author provide ample detail describing what has been done, stakeholders, and what can be done to improve the process in the future. Minor comments are below:
- It is often unclear if the authors were describing what was done as part of this initiative or efforts that were done previously or outside of this particular initiative.
- It has been a few years since this initiative has started, have any data been collected and analyzed? It would be interesting to describe any trends in reporting, collecting, or documenting WRI's that may have been observed post-implementation or if increasing awareness has resulted in any changes.
Note: I could not see Figure 1 as it was very blurry, so cannot provide any comments related to this figure.
Author Response
Dear Reviewer 2,
Thank you for your thoughtful recommendations and constructive feedback, they are much appreciated and all noted.
We provide a point by point response to your review below:
Comments and Suggestions
- The authors describe ongoing efforts made in Qatar to address increasing work-related injuries and what was learned from implementing these efforts. The author provide ample detail describing what has been done, stakeholders, and what can be done to improve the process in the future. Minor comments are below:
- It is often unclear if the authors were describing what was done as part of this initiative or efforts that were done previously or outside of this particular initiative.
- Duly noted, we have added a clarificatory statement at the start of Section 2 to delineate this point.
- It has been a few years since this initiative has started, have any data been collected and analyzed? It would be interesting to describe any trends in reporting, collecting, or documenting WRI's that may have been observed post-implementation or if increasing awareness has resulted in any changes.
- Duly noted, we have added a statement, at the end of Section 2.5, to describe this point. The ILO is supporting the continued data collection of this unified registry and we hope to analyze and report this data within a year.
- It is often unclear if the authors were describing what was done as part of this initiative or efforts that were done previously or outside of this particular initiative.
Best regards,
WURQ Team
Reviewer 3 Report
The paper contains an overview of the work-related injuries (WRI) in Qatar and details the efforts to address WRI using a public health approach. The authors provided a wide description of contextual factors and multidisciplinary efforts for the prevention of WRI. Also, the paper described the results of a prevention program supported by a National Research Priorities Program. In this perspective, the authors aimed to implement a public health approach to achieve long-term control in Qatar.
The English language is sufficiently appropriate and understandable.
I think that the paper is significant for the field and of interest to a general audience. The authors provided an advance in current knowledge of the WRI.
However, some weaknesses in the paper should be address by the authors. Hence, I include the following considerations about the manuscript in its current version:
- Introduction: I am not sure to understand how the introduction is linked with the other sections of the manuscript. I think this could decrease the readability of the paper.
- Figure 1 is unreadable.
- Introduction and discussion: I have some doubts about the heterogeneity of factors examined. In light of this perspective, a more careful look at psychological factors involved in WRI is necessary. Protective and risk factors (i.e., burnout and resilience) could influence the WRI. So, these papers should be analysed and included by the authors:
Lenzo, V., Bordino, V., Bonanno, G. A., & Quattropani, M. C. (2020). Understanding the role of regulatory flexibility and context sensitivity in preventing burnout in a palliative home care team. PLoS ONE, 15(5),e0233173.
Lenzo, V., Maisano, G., Garipoli, C., Aragona, M., Filastro, A., Verrastro, V., Petralia, M. C., & Quattropani, M. C. (2020). Risk of burnout in a sample of oncology healthcare professionals working in a hospital oncology unit with hospice and relationship with dysfunctional metacognitive beliefs. Minerva Psichiatrica, 61(1).
My recommendation for the Editor is to accept after minor revision.
Author Response
Dear Reviewer 3,
Thank you for your thoughtful recommendations and constructive feedback, they are much appreciated, and all noted.
We provide a point by point response to your review below:
Comments and Suggestions
- The paper contains an overview of the work-related injuries (WRI) in Qatar and details the efforts to address WRI using a public health approach. The authors provided a wide description of contextual factors and multidisciplinary efforts for the prevention WRI. Also, the paper described the results of a prevention program supported by a National Research Priorities Program. In this perspective, the authors aimed to implement a public health approach to achieve long-term control in Qatar.
- The English language is sufficiently appropriate and understandable.
- I think that the paper is significant for the field and of interest to a general audience. The authors provided an advance in current knowledge of the WRI. However, some weaknesses in the paper should be address by the authors. Hence, I include the following considerations about the manuscript in its current version:
- - Introduction: I am not sure to understand how the introduction is linked with the other sections of the manuscript. I think this could decrease the readability of the paper.
- We realize that many readers will not be well-versed or conversant with the occupational safety and demographic context of Qatar so we included this abbreviated description in the introduction. If the Editorial team feels it is superfluous then we will remove it from the manuscript.
- - Figure 1 is unreadable
- Duly noted, we have provided higher resolution figures.
- Introduction and discussion: I have some doubts about the heterogeneity of factors examined. In light of this perspective, a more careful look at psychological factors involved in WRI is necessary. Protective and risk factors (i.e., burnout and resilience) could influence the WRI. So, these papers should be analysed and included by the authors:
- Lenzo, V., Bordino, V., Bonanno, G. A., & Quattropani, M. C. (2020). Understanding the role of regulatory flexibility and context sensitivity in preventing burnout in a palliative home care team. PLoS ONE, 15(5),e0233173.
- Lenzo, V., Maisano, G., Garipoli, C., Aragona, M., Filastro, A., Verrastro, V., Petralia, M. C., & Quattropani, M. C. (2020). Risk of burnout in a sample of oncology healthcare professionals working in a hospital oncology unit with hospice and relationship with dysfunctional metacognitive beliefs. Minerva Psichiatrica, 61(1).
- Thank you for reminding us of these important factors that should be included in this report, we have added 2 sentences acknowledging these in the Limitations section, as recommended. We however cited these 4 references that we felt were more focused on these issues for migrant workers, in lieu of your recommended references.
- Fernández-Esquer ME, Gallardo KR, Diamond PM. Predicting the Influence of Situational and Immigration Stress on Latino Day Laborers' Workplace Injuries: An Exploratory Structural Equation Model. J Immigr Minor Health. 2019;21(2):364-371. doi:10.1007/s10903-018-0752-3
- Ramos AK, Carlo G, Grant K, Trinidad N, Correa A. Stress, Depression, and Occupational Injury among Migrant Farmworkers in Nebraska. Safety (Basel). 2016;2(4):23. doi:10.3390/safety2040023
- Qiu P, Caine E, Yang Y, Chen Q, Li J, Ma X. Depression and associated factors in internal migrant workers in China. J Affect Disord. 2011;134(1-3):198-207. doi:10.1016/j.jad.2011.05.043
- Sweileh WM. Global output of research on the health of international migrant workers from 2000 to 2017. Global Health. 2018;14(1):105. Published 2018 Nov 8. doi:10.1186/s12992-018-0419-9.
- Thank you for reminding us of these important factors that should be included in this report, we have added 2 sentences acknowledging these in the Limitations section, as recommended. We however cited these 4 references that we felt were more focused on these issues for migrant workers, in lieu of your recommended references.
- - Introduction: I am not sure to understand how the introduction is linked with the other sections of the manuscript. I think this could decrease the readability of the paper.
Best regards,
WURQ Team
Reviewer 4 Report
The manuscript described the circumstances surrounding work-related injuries in Qatar, featuring migrant workers. Although there is no quantitative analysis with the statistical method, the contents are interesting and of considerable importance deserve to be published.
The quality of the figures, however, is poor.The characters used in Figure 1 are unreadable due to poor resolution.
Figure 2 may be easier to understand if it is explained a little more in the figure legend or in the text, especially for the flow of foreign workers.
Author Response
Dear Reviewer 4,
Thank you for your thoughtful recommendations and constructive feedback, they are much appreciated and all noted.
We provide a point by point response to your review below:
Comments and Suggestions
- The manuscript described the circumstances surrounding work related injuries in Qatar, featuring migrant workers. Although there is no quantitative analysis with the statistical method, the contents are interesting and of considerable importance deserve to be published.
- Thank you for these comments.
- The quality of the figures, however, is poor
- Duly noted, we have provided higher resolution figures.
- The characters used in Figure 1 are unreadable due to poor resolution.
- Duly noted, we have provided higher resolution figures.
- Figure 2 may be easier to understand if it is explained a little more in the figure legend or in the text, especially for the flow of foreign workers.
- Duly noted, we have provided higher resolution figures.
Best regards,
WURQ Team
Round 2
Reviewer 1 Report
You still need to review:
The knowledge gap indicated does not support the objectives of the study.
Although relevant, the resolution of the figure is still low and with blurred letters.
Author Response
Dear Reviewer 1,
We apologize for not thoroughly addressing your initial review. We had an internal discussion that revolved around providing too much of a political commentary and a more academic description of the ‘gap’. In the end, thanks to your efforts to ‘pin us down’ [so to speak] we are providing the manuscript with a compromise description. It is our hope that this is an acceptable description of the evidence gap that corresponds with the objectives of the paper.
We provide a point by point response to your review below:
Comments and Suggestions for Authors
You still need to review:
The knowledge gap indicated does not support the objectives of the study.
We have made revisions in lines 49-54 in the revised manuscript.
We look forward to hearing back from you.
Best regards,
WURQ Team